# EgoTracks: A Long-term Egocentric Visual Object Tracking Dataset

**Hao Tang**[1], **Kevin J Liang**[1], **Kristen Grauman**[1,2], **Matt Feiszli**[1,*], **Weiyao Wang**[1,*]

FAIR, Meta[1], UT Austin[2], Equal Contribution[*]

{haotang, kevinjliang, grauman, mdf, weiyaowang}@meta.com

## Abstract

Visual object tracking is key to many egocentric vision problems. However, the full spectrum of challenges of egocentric tracking faced by an embodied AI is under-represented in many existing datasets, which tend to focus on short, third-person videos. Egocentric video has several distinguishing characteristics from those commonly found in past datasets: frequent large camera motions and hand interactions with objects commonly lead to occlusions or objects exiting the frame, and object appearance can change rapidly due to widely different points of view, scale, or object states. Embodied tracking is also naturally long-term, and being able to consistently (re-)associate objects to their appearances and disappearances over as long as a lifetime is critical. Previous datasets under-emphasize this re-detection problem, and their "framed" nature has led to adoption of various spatiotemporal priors that we find do not necessarily generalize to egocentric video. We thus introduce EgoTracks, a new dataset for long-term egocentric visual object tracking. Sourced from the Ego4D dataset, EgoTracks presents a significant challenge to recent state-of-the-art single-object trackers, which we find score more poorly on our new dataset than existing popular benchmarks, according to traditional tracking metrics. We further show improvements that can be made to the STARK tracker to significantly increase its performance on egocentric data, resulting in a baseline model we call EgoSTARK. We publicly release our annotations and benchmark (`https://github.com/EGO4D/episodic-memory/tree/main/EgoTracks`), hoping our dataset leads to further advancements in tracking.

## 1 Introduction

First-person or "egocentric" computer vision aims to capture the real-world perceptual problems faced by an embodied AI; it has drawn strong recent interest as an underserved but highly relevant domain of vision, with important applications ranging from robotics [67, 18] to augmented and mixed reality [2, 70, 29]. Visual object tracking (VOT), long a fundamental problem in vision, is a core component to many egocentric tasks, including tracking the progress of an action or activity, (re-)association of objects in one's surroundings, and predicting future states of the environment. Yet, while the VOT field has made many significant advancements over the past decade, tracking in egocentric video remains underexplored. This lack of attention is in large part due to the absence of a large-scale egocentric tracking dataset for training and evaluation. While the community has proposed a number of popular tracking datasets in recent years, including OTB [81], TrackingNet [60], GOT-10k [33], and LaSOT [21], we find that the strong performance that state-of-the-art trackers achieve on these benchmarks does not translate well to egocentric video, thus establishing a strong need for such a tracking dataset.

We attribute this performance gap to the many unique aspects of egocentric views compared to the more traditional third-person views of previous datasets. In contrast to intentionally "framed" video, egocentric videos are often uncurated, meaning they tend to capture many attention shifts between

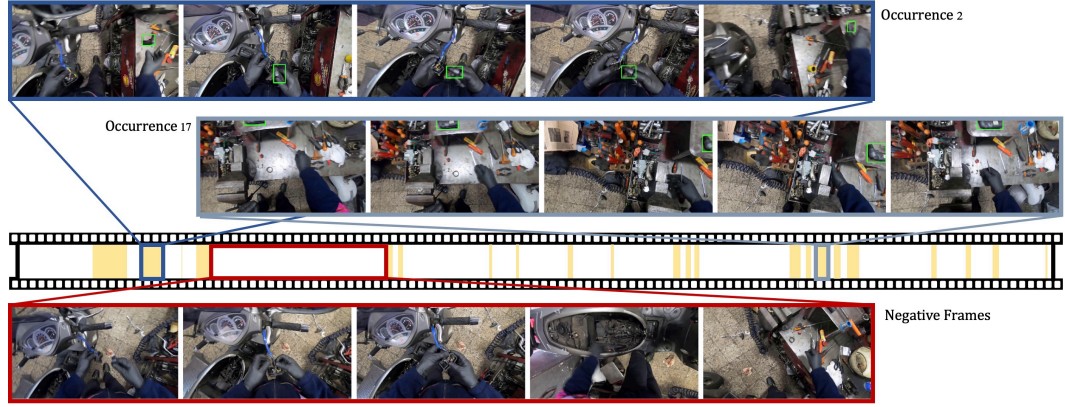

Figure 1: A video from the proposed EgoTracks dataset, with yellow clip segments marking when the object (`blowtorch`) is visible. Note the frequent disappearances and reappearances of the object over an 8 minute video, with lengthy absences, necessitating re-detection to track accurately without false positives. The egocentric nature of the video includes the camera-wearer interacting with the object (Occurrence 2), resulting in significant hand occlusions and dramatic changes in pose.

activities, objects, or locations. Due to the first-person perspective, large head motions from the camera wearer often result in objects repeatedly leaving and re-entering the field of view; similarly, hand manipulations of objects [68] leads to frequent occlusions, rapid variations in scale and pose, and potential changes in state or appearance. Furthermore, egocentric video tends to be long (sometimes representing the entire life of an agent or individual), meaning the volume of the aforementioned occlusions and transformations scales similarly. These characteristics all make tracking objects in egocentric views dramatically more difficult than scenarios commonly considered in prior datasets, and their absence represents an evaluation blindspot.

Head motions, locomotion, hand occlusions, and temporal length lead to several challenges. First, frequent object disappearances and reappearances causes the problem of *re-detection* within egocentric tracking to become especially critical. Many previous tracking datasets primarily focus on short-term tracking in third-person videos, providing limited ability to evaluate many of the challenges of long-term egocentric tracking due to low quantity and length of target object disappearances. As a result, competent re-detection is not required for strong performance, leading many recent short-term trackers to neglect it, instead predicting a bounding box for every frame, which may lead to rampant false positives or tracking the wrong object. Additionally, the characteristics of short-term third-person video have also induced designs relying on gradual changes in motion and appearance. As we later show (Section 5.2), many of the motion, context, and scale priors made by previous short-term tracking algorithms fail to transfer to egocentric video.

Notably, re-detection, occlusions, and longer-term tracking have long been recognized as difficult for VOT as a field, leading to recent benchmark construction efforts [54, 11, 58, 73, 34, 75] emphasizing these aspects. We argue that egocentric video provides a natural source for these challenges at scale while also representing a highly impactful application for tracking, therefore constituting a significant opportunity. We thus present **EgoTracks**: a large-scale long-term egocentric visual object tracking dataset for training and evaluating long-term trackers. Seeking a realistic challenge, we source videos from Ego4D [29], a large-scale dataset consisting of unscripted, in-the-wild egocentric videos of daily-life activities. The result is a large-scale dataset to evaluate the tracking and re-detection ability of SOT models, with more than 22,028 tracks from 5708 average 6-minute videos. This

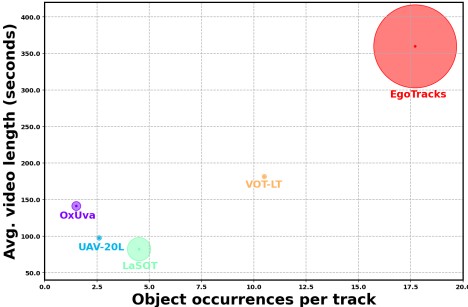

Figure 2: EgoTracks is an order of magnitude larger than past long-term VOT datasets, with significantly more tracks (circle area) and object disappearances/appearances in longer videos.

constitutes the first large-scale dataset for visual object tracking in egocentric videos in diverse settings, providing a new, significant challenge compared with previous datasets.

We perform a thorough analysis of our new dataset and its new characteristics relative to prior benchmarks, demonstrating its difficulty and the need for further research to develop trackers capable of handling long-term egocentric vision. Our experiments reveal remaining open problems and insights towards promising directions in egocentric tracking. Leveraging these intuitions, we propose multiple simple yet effective changes, such as adjustment of spatiotemporal priors, egocentric data finetuning, and combining multiple templates. We apply these proposed strategies on a state-of-the-art (SOTA) STARK tracker [84], training a strong tracker dedicated to long-term egocentric tracking: **EgoSTARK**. We hope Ego-STARK can serve as a strong baseline and facilitate future research.

We make the following contributions:

1. We present EgoTracks, the first large-scale long-term object tracking dataset with diverse egocentric scenarios. We analyze its uniqueness in terms of evaluating the re-detection performance of trackers.

2. We conduct comprehensive experiments to understand the performance of many state-of-the-art trackers on the EgoTracks validation set and observe that due to the biases and evaluation blindspots of existing third-person datasets, they tend to struggle.

3. We perform an analysis of what makes a good tracker for long-form egocentric video. Applying these learnings to the STARK tracker [84], we produce a strong baseline we call EgoSTARK, which achieves significant improvements (+15% F-score) on EgoTracks.

## 2 Related work

### 2.1 Visual object tracking datasets

Visual object tracking studies the joint spatial-temporal localization of objects in videos. From a video and a predefined taxonomy, multiple object tracking (MOT) models simultaneously detect, recognize, and track multiple objects. For example, MOT [57] tracks humans, KITTI [26, 53] tracks pedestrians and cars, and TAO [15] tracks a large taxonomy of 833 categories. In contrast, single object tracking (SOT) follows a single object via a provided initial template of the object, without any detection or recognition. Thus, SOT is often taxonomy-free and operates on generic objects. The community has constructed multiple popular benchmarks to study this problem, including OTB [81], UAV [59], NfS [38], TC-128 [48], NUS-PRO [43], GOT-10k [33], VOT [40], and TrackingNet [60].

While these SOT datasets mainly consist of short videos (e.g. a few seconds), long-term tracking has been increasingly of interest. Tracking objects in longer videos (several minutes or more) poses unique challenges, e.g. significant transformations, displacements, disappearances, and reappearances. On top of localizing the object when visible, the model also must produce no box when the object is absent, and then re-localize the same object when it reappears. OxUvA [73] is one of the first to benchmark longer videos (average 2 minutes), with 366 evaluation-only videos. LaSOT [21] scales this to 1400 videos with more frequent object reappearances. Concurrently, VOT-LT [39] includes frequent object disappearances and reappearances in 50 purposefully selected videos.

Our EgoTracks focuses on long-term SOT and presents multiple critical and unique attributes: 1) significantly larger scale, with **5708** videos of an average **6 minutes** (Figure 2); 2) more frequent disappearances & reappearances (avg. **17.7** times) happening in natural, real-world scenarios; 3) data sourced from egocentric videos shot in-the-wild, involving unique challenging situations, such as large camera motions, diverse perspective changes, hand-object interactions, and frequent occlusions.

### 2.2 Single object tracking methodologies

Many modern approaches use convolutional neural networks (CNNs), either with Siamese network [45, 76, 44] or correlation-filter based [13, 3, 8, 56, 4] architectures. With recent successes in vision tasks like classification [17] and detection [5], Transformer architecture [74] for tracking have also become popular. For example, TransT [6] uses attention-based feature fusion to combine features of the object template and search image. More recently, several works utilize Transformers as direct predictors to achieve a new state of the art, such as STARK [84], ToMP [55] and SBT [82]. These models tokenize frame features from a ResNet [31] encoder, and use a Transformer to predict the bounding box and object presence score with the feature tokens. These methods are often developed on short-term SOT datasets and assume that the target object stays in the field of view with minimum occlusions. On the other hand, long-term trackers [75, 34, 11] are designed to cope with

Table 1: **Object tracking datasets comparison.** In addition to larger scale than previous datasets, the scenarios captured by EgoTracks represent a significantly harder challenge for SOTA trackers, suggesting room for improved tracking methodology.

| Dataset | Video Hours | Avg. Length (s) | Ann. FPS | Ann. Type | Egocentric | SOTA (P/AO)* |
|---|---|---|---|---|---|---|
| ImageNet-Vid [66] | 15.6 | 10.6 | 25 | mask | No | |
| YT-VOS [83] | 5.8 | 4.6 | 5 | mask | No | -/83.6 [32] |
| DAVIS 17 [64] | 0.125 | 3 | 24 | mask | No | -/86.3 [7] |
| TAO [15] | 29.7 | 36.8 | 1 | mask | No | |
| UVO [79] | 2.8 | 10 | 30 | mask | No | -/73.7 [61] |
| EPIC-KITCHENS VISOR [14] | 36 | 12** | 0.9 | mask | **Yes** | -/74.2 [61] |
| GOT-10k [33] | 32.8 | 12.2 | 10 | bbox | No | -/75.6 [9] |
| OxUvA [73] | 14.4 | 141.2 | 1 | bbox | No | |
| LaSOT [21] | 31.92 | 82.1 | 30 | bbox | No | 80.3/- [9] |
| TrackingNet [60] | 125.1 | 14.7 | 28 | bbox | No | 86/- [9] |
| TREK-150 [19, 20] | 0.45 | 10.81 | 60 | bbox | **Yes** | -/50.5 [19, 20] |
| **EgoTracks (Ours)** | **602.9** | **367.9** | 5 | bbox | **Yes** | 45/54.1 |

*: P: Precision, AO: average overlap (J-Score for mask-based datasets). **: Original video is 720s.

the problem of re-detecting objects in their reappearances. Designed to be aware of potential object disappearances, these approaches search the whole image for its reappearance.

## 2.3 Tracking in egocentric videos

Multiple egocentric video datasets have been introduced in the past decades [12, 29, 41, 71, 63, 23], offering a host of interesting challenges, many of which require associating objects across frames: activity recognition [36, 46, 85, 80, 27], anticipation [22, 24, 28], video summarization [16, 41, 42, 52], human-object interaction [14, 50], episodic memory [29], visual query [29], and camera-wearer pose inference [35]. To tackle these challenges, tracking is leveraged in many methodologies [29, 14, 51, 42, 50], yet few works have been dedicated to this fundamental problem on its own. [19, 20] have started to recognize the challenges of egocentric object tracking and might be the most related work to ours. The major difference, however, is the scale of the dataset: [19, 20] contain 150 tracks intended only for evaluation, while EgoTracks is 100× larger (see Table 1), containing 20k tracks with training and evaluation splits. Also, while past efforts have sourced videos from the kitchen-heavy EPIC-KITCHEN [12], EgoTracks sources videos from Ego4D [29], which has more diverse scenarios. EgoTracks provides a unique, large-scale testbed for developing tracking methods dedicated to egocentric videos; our improved baseline EgoSTARK also serves as a potential plug-and-play module to solve other tasks where object association is desired.

In egocentric video understanding, Ego4D [29] and EPIC-KITCHENS VISOR [14] are closely related. Ego4D contains the largest collection of egocentric videos in-the-wild; EgoTracks is annotated on a subset of Ego4D. In addition, Ego4D proposes many novel tasks, such as Episodic Memory, with tracking identified as a core component. VISOR was introduced concurrently, annotating short-term (12 sec on average) videos from EPIC-KITCHENS [12] with instance segmentation masks. We believe EgoTracks offers multiple unique values complementary to EPIC-VISOR: long-term tracking (6 min vs. 12 sec), significantly larger-scale (5708 video clips vs. 158), and more diversified video sources (80+ scenes vs. kitchen-only; see Figure 4).

## 3 The EgoTracks dataset

We present EgoTracks: a large-scale long-term egocentric single object tracking dataset, consisting of a total of 22028 tracks from 5708 videos. We follow the same data split as the Ego4D Visual Queries (VQ) 2D benchmark (see Appendix A for details).

### 3.1 Ego4D visual queries (VQ) benchmark

Ego4D [29] is a massive-scale egocentric video dataset, consisting of 3670 hours of diverse daily-life activities of consenting participants in an in-the-wild format; the videos have personally identifiable information removed and were screened for offensive content. The dataset is accompanied by multiple benchmarks, but the most relevant task for our purposes is episodic memory's 2D VQ task: Given an egocentric video and a cropped image of an object, the goal is to localize when and where the object was last seen in the video, as a series of 2D bounding boxes in consecutive frames. This task is closely related to long-term tracking: finding an object in a video given a visual template is identical

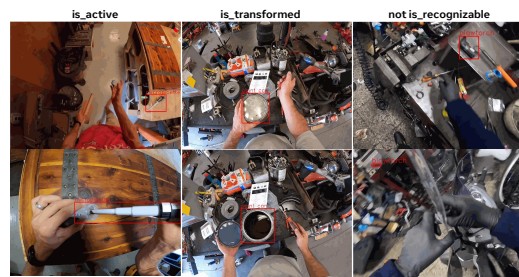

Figure 3: **EgoTracks tracklet attribute examples.** *Left*: Micropipette on a bench (top) versus actively used (bottom). *Middle*: A paint can (top) is opened (bottom). *Right*: A blowtorch (top) requiring context from other frames to identify due to distance and motion blur (bottom).

Table 2: **Track attributes** in train/val sets.

|  | Total number | Percentage |
|---|---|---|
| All Tracks | 17593 | 100% |
| is_active | 3963 | 22.52% |
| is_transformed | 1080 | 6.13% |
| is_recognizable | 17557 | 99.79% |

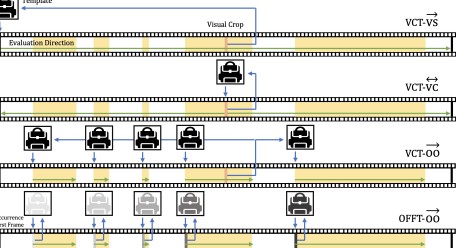

Table 3: Evaluation protocols visualization.

to the re-detection problem in long-term tracking. Moreover, Ego4D's baselines rely heavily on tracking methods: Siam-RCNN [75] and KYS [4] for global and local tracking, respectively.

**Shortcomings.** While highly related, the VQ dataset is not immediately suitable long-term tracking. In particular, the VQ annotation guidelines were roughly the following: 1) identify three different objects that appear multiple times in the video; 2) annotate a query template for each object, which should contain the entire object without any motion blur; 3) annotate an occurrence of the object that is temporally distant from the template. Thus, these annotations are not exhaustive over time (they are quite sparse), limiting their applicability to tracking. On the other hand, the selection criteria result in a strong set of candidate objects, which we leverage to build EgoTracks.

### 3.2 Annotating VQ for long-term tracking

We thus start with the VQ visual crop and response track, asking annotators to first identify the object represented by the visual crop, the response track, and object name. From the video's start, we instruct the annotators to draw a bounding box around the object each time it appears. Because annotators must go through each video in its entirety, which contain an average of ∼1800 frames at 5 frames per second (FPS), this annotation task is labor-intensive, taking roughly 1 to 2 hours per track. An important aspect of this annotation is its exhaustiveness: the entire video is densely annotated for the target object, and any frame without a bounding box is considered as a negative. Being able to reject negatives examples is an important component to re-detection in real-world settings, as false positives can impact certain applications as much as false negatives.

**Quality Assurance.** All tracks are quality checked by expert annotators after the initial annotations. To measure the annotation quality, we employ multi-review on a subset of the validation set. Three independent reviewers are asked to annotate the same video. We find the overlaps between these independent annotations are high ($> 0.88$ IoU). Further, since EgoTracks has a focus on re-detection, we check the temporal overlap of object presence and find it to be consistent across annotators. In total, the entire annotation effort represented roughly 86k worker-hours of effort.

### 3.3 Tracklet attributes

In addition to the bounding box annotations, we also label certain relevant attributes to allow for different training strategies or deeper analysis of validation set performance. We annotate the following three attributes per occurrence (see Figure 3 for examples and Table 2 for statistics):

- is_active: In Ego4D, the camera wearer often interacts with relevant objects with their hands. Objects in the state of being handled pose a challenge for tracking algorithms due to frequent occlusion and rapid changes in pose.
- is_transformed: Objects in Ego4D may undergo transformations, such as deformations and state changes. Such instances require being able to quickly adapt to the tracked object having a new appearance.
- is_recognizable: Due to occlusions, motion blur, scale, or other conditions, some objects in Ego4D can be extremely difficult to recognize without additional context. We thus annotate if the object is recognizable solely based on its appearance, without using additional context information (*e.g.* other frames).

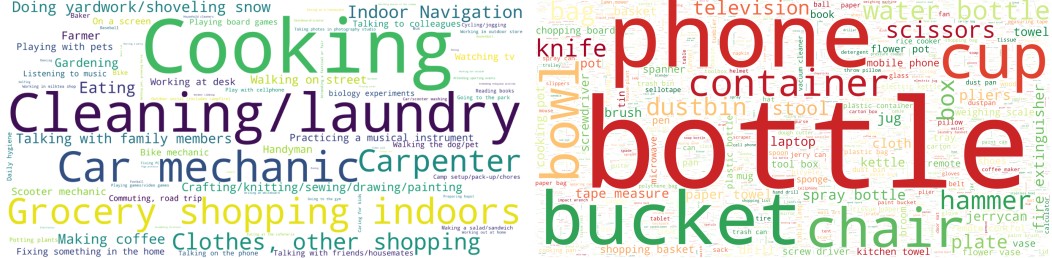

Figure 4: EgoTracks is a large-scale egocentric dataset of diverse scenarios (left) and objects (right).

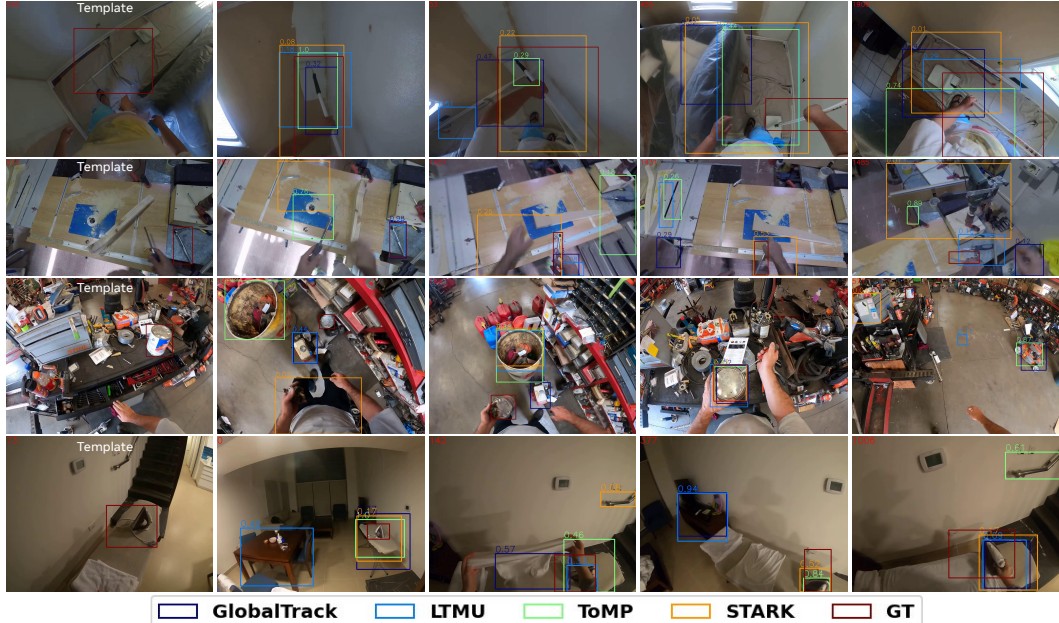

Figure 5: Qualitative results of different trackers. EgoTracks presents significant challenges for all trackers, due to drastic viewpoint changes, occlusions, changes in scale, head motion etc.

## 3.4 Data diversity

Figure 4 shows the diversity of EgoTracks in terms of both the actions the videos contain and name of the object tracks. Ego4D [29] videos depict a wide variety of in-the-wild activities (Figure 4, left), captured in locations all around the world. As such, they represent a far more diverse set of scenes than other egocentric datasets [12], thus requiring models to learn more general representations.

There was no fixed taxonomy for which objects to track. Instead, annotators were asked to choose objects that are "interesting" [29], which were named freely, resulting in considerable object diversity. We extracted the nouns from these object names, resulting in around 1000+ unique names. Figure 4, right shows the wide array of objects tracked in EgoTracks.

## 4 Analysis of state-of-the-art SOT trackers

We compare the performance of several off-the-shelf tracking models on EgoTracks's validation set. Identifying STARK [84] as the one with the best performance, we conduct further ablation studies under different settings using STARK to further understand its behavior.

### 4.1 Evaluation protocols and metrics

**Evaluation Protocols.** We introduce several evaluation protocols for EgoTracks, consisting of different combinations of the initial template, evaluated frames, and the temporal direction in which the tracker is run. For the initial template, we consider two choices:

Table 4: **EgoTracks performance comparison.** Off-the-shelf, all trackers perform poorly, demonstrating the new challenges of EgoTracks. Higher performance from tracking by detection methods + Oracle imply that instance association, not detection, is one of the primary challenges.

| Method | AO | F-score | Precision | Recall | FPS |
|---|---|---|---|---|---|
| KYS [4] | 16.09 | 13.09 | 12.50 | 13.74 | 20 |
| DiMP [3] | 16.45 | 11.84 | 10.31 | 13.91 | 43 |
| GlobalTrack [34] † | 23.63 | 20.40 | 31.28 | 15.14 | 6 |
| LTMU [11] † | 29.33 | 27.46 | 37.28 | 21.74 | 13 |
| ToMP [55] | 30.93 | 20.95 | 19.63 | 22.46 | 24.8 |
| Siam-RCNN [75] † | 37.48 | 35.38 | 52.80 | 26.67 | 4.7 |
| MixFormer (MixViT-L, ConvMAE) [9, 10] | 27.93 | 25.54 | 28.30 | 23.27 | 10 |
| STARK [84] - Res50 | 35.99 | 30.48 | 34.70 | 27.17 | 41.8 |
| STARK [84] - Res101 | 35.03 | 30.18 | 35.30 | 26.35 | 31.7 |
| **Tracking by Detection** † | | | | | |
| Mask R-CNN [30]+Oracle | 60.00 | - | - | - | |
| GGN [78]+Oracle | 75.92 | - | - | - | |
| GGN+InstEmb | 15.19 | 9.92 | 11.75 | 8.58 | |

†: trackers with re-detection

- **Visual Crop Template (VCT)**: The visual crop images were specifically chosen to be high-quality views of the target and served as our annotators' references for identifying the object throughout the videos. Thus, they make ideal candidates for initializing a tracker.
- **Occurrence First Frame Template (OFFT)**: The tracker is initialized with the first frame of each occurrence (see $\overrightarrow{OO}$ below). While this may result in a lower quality view of the object, temporal proximity to subsequent frames means it may be closer in appearance.

Note that we exclude the template frame from the calculation of any evaluation metrics. We also consider several choices for the evaluated frames and temporal direction:

- **Video Start Forward ($\overrightarrow{VS}$)**: The tracker is evaluated on every frame of the video in causal order, starting from the first frame. This represents a tracker's ability to follow an object through a long video.
- **Visual Crop Forward/Backward ($\overleftrightarrow{VC}$)**: The tracker is run on the video twice, once starting at the visual crop frame and running forward and time, and a second time running backwards. This represents an alternative way of covering every frame in the video, but with closer visual similarity between **VCT** initialization and the first frames encountered by the tracker.
- **Occurrences Only Forward ($\overrightarrow{OO}$)**: The tracker is only evaluated on the object occurrences, when the object is visible. This simplifies the tracking task and allows us to dis-entangle the challenge of re-detection from that of simply tracking in an egocentric clip.

We specify protocols by concatenating the appropriate descriptors. We primarily consider **VCT-$\overrightarrow{VS}$**, **VCT-$\overleftrightarrow{VC}$**, **VCT-$\overrightarrow{OO}$**, and **OFFT-$\overrightarrow{OO}$** (Table 3) in our experiments.

**Metrics.** We adopt common metrics in object tracking, including F-score, precision, and recall; details can be found in [54]. Trackers are ranked mainly by the F-score. We additionally consider average overlap (AO), success, precision, and normalized precision as short-term tracking metrics [72].

## 4.2 SOT trackers struggle on EgoTracks

We compare the performance of select trackers on EgoTracks with the **VCT-$\overrightarrow{VS}$** evaluation protocol. Given the breadth of tracking algorithms, we do not aim to be exhaustive but select high-performing representatives of different tracking principles. KYS [4] and DiMP [3] are two short-term tracking algorithms that maintain an online target representation. ToMP [55], STARK [84] and MixFormer [9, 10] are three examples of the SOTA short-term trackers based on Transformers. GlobalTrack [34] is a global tracker that searches the entire search image for re-detection. LTMU [11] is a high performance long-term tracker that combines a global tracker (GlobalTrack) with a local tracker (DiMP). Siam R-CNN [75] leverages dynamic programming to model a full path of history for long-term. The performance of these trackers on EgoTracks are summarized in Table 4. AO in this table is equivalent to recall at the probability threshold 0. Qualitative results are shown in Figure 5.

Table 5: Comparing tracker initializations. (Left) Comparison of trackers initialized from the first frame in each occurrence and tracking only that single occurrence (oracle re-detection). (Right) STARK whole-video performance, starting from video start frame vs. the visual crop frame.

| Method | AO | Success | Pre | $\text{Pre}_{\text{norm}}$ |
|---|---|---|---|---|
| KYS [4] | 33.92 | 34.87 | 31.22 | 34.87 |
| DiMP [3] | 34.80 | 35.70 | 32.13 | 38.98 |
| ToMP [55] | 45.17 | 45.93 | 41.74 | 47.88 |
| STARK [84] | 50.01 | 50.64 | 45.76 | 51.91 |

| Method | AO | F-score | Precision | Recall |
|---|---|---|---|---|
| STARK - **VCT-$\overrightarrow{\text{VS}}$** | 35.99 | 30.48 | 34.70 | 27.17 |
| STARK - **VCT-$\overleftrightarrow{\text{VC}}$** | 40.01 | 34.02 | 38.31 | 30.60 |

Table 6: **OFFT-$\overrightarrow{\text{OO}}$** AO of standard STARK model [84] for each attribute.

| Attribute | True | False |
|---|---|---|
| is_active | 49.65 | 55.73 |
| is_transformed | 49.19 | 55.31 |
| is_recognizable | 55.52 | 46.65 |

Table 7: Performance of trackers finetuned on EgoTracks.

| Method | AO | F-score | Precision | Recall |
|---|---|---|---|---|
| ToMP | 36.13 | 28.11 | 29.01 | 27.26 |
| Siam-RCNN | 45.67 | 41.41 | 56.11 | 32.81 |
| STARK | 44.25 | 38.20 | 42.06 | 34.99 |

We highlight several observations. First, the object presence scores from most short-term trackers are not very useful, as can be seen from the low precision of KYS (12.5), DiMP (13.91), and ToMP (19.63), while long-term trackers like GlobalTrack, DiMP_LTMU and Siam R-CNN achieve higher precisions at 31.28, 37.28 and 52.8. This is expected as long-term trackers are designed to place more emphasis on high re-detection accuracy, though there clearly is still room for improvement. STARK achieves the second highest precision at 34.70, which is an exception as it has a second training stage to teach the model to classify whether the object is present. Second, more recent works such as MixFormer and STARK achieve better F-score than previous short-term trackers. This could be partially due to advances in training strategies, more data, and Transformer-based architectures. Surprisingly, we found recent MixFormer [10] does not outperform STARK, despite achieving new SOTA on its training dataset. This highlights a potential difficulty in generalization.

We also include results using the principle of Tracking by Detection [62, 1]: a detector proposes 100 bounding boxes, and we select the best using cosine similarity of box features. We observe that an open-world detector GGN [78] trained on COCO [49] generalize reasonably well with oracle matching, achieving 75.92 AO. However, the association problem is very challenging, bringing down the AO to 15.19. Implementation details can be found in Appendix B.

### 4.3 Re-detection and diverse views are challenging

We perform additional EgoTracks experiments following alternative evaluation protocols to gain further insights on tracker performance (Table 5). To decouple the re-detection problem from other egocentric aspects of EgoTracks, we evaluate with the **OFFT-$\overrightarrow{\text{OO}}$** protocol, which ignores the negative frames of the video, thus obviating the need for re-detection. Unsurprisingly, all trackers do significantly better, emphasizing the challenging nature of re-detection in EgoTracks. We also run experiments in the **VCT-$\overleftrightarrow{\text{VC}}$** setting, where the initial template is temporally adjacent to the first tracked frames. Here we see a 3-4% improvement to AO, F-score, precision, and recall compared to the **VCT-$\overrightarrow{\text{VS}}$** protocol, illustrating that trackers like STARK are designed to expect gradual transitions in appearance. Both these experiments illustrate that the re-detection problem is a significant challenge for tracking and the need for better long-term benchmarks.

### 4.4 Attributes capture hard scenarios for tracking

We use the validation set tracklet attribute annotations described in Section 3.3 to further understand performance on our evaluation set. For each attribute, we split the tracklets into two groups, corresponding to the attribute being true and false. We then use a standard STARK tracker [84] and report AO for each group of tracklets using the **OFFT-$\overrightarrow{\text{OO}}$** evaluation protocol in Table 6. As might be expected, we find that when objects are being actively used by the user or in the midst of a transformation, AO tends to be lower, by roughly 6%, likely due to occlusions or changes in appearance. Additionally, STARK tends to have a harder time when the object is hard to recognize in the image, whether due to occlusions, blur, scale, or other conditions.

Table 8: Evaluating EgoTracks on EPIC-VISOR [14].

| Method | AO | F1 | Precision | Recall |
|---|---|---|---|---|
| Siam-RCNN | 51.0 | 42.4 | 37.7 | 48.4 |
| Siam-RCNN (finetuned on EgoTracks) | 52.7 | 43.5 | 38.4 | 50.3 |
| STARK | 59.2 | 46.4 | 39.4 | 56.6 |
| STARK (finetuned on EgoTracks) | 61.4 | 48.5 | 41.0 | 59.4 |
| EgoSTARK | 64.7 | 50.6 | 42.6 | 62.3 |

Table 9: Train/test-time hyperparameters comparison.

| | Method | AO | F-score | Precision | Recall |
|---|---|---|---|---|---|
| Data | STARK | 35.99 | 30.48 | 34.70 | 27.17 |
| | STARK - ft on VQ | 38.94 | 33.53 | 39.13 | 29.33 |
| | STARK - ft on EgoTracks | 44.25 | 38.20 | 42.06 | 34.99 |
| Augmentation | STARK - ft on VQ | 38.94 | 33.53 | 39.13 | 29.33 |
| | STARK - ft + multiscale | 48.44 | 41.92 | 42.65 | 41.30 |
| Search window | search_size = 320 | 35.99 | 30.48 | 34.70 | 27.17 |
| | search_size = 480 | 48.21 | 39.69 | 43.95 | 36.19 |
| | search_size = 640 | 52.09 | 42.39 | 46.23 | 39.15 |
| | search_size = 800 | 54.08 | 43.74 | 47.60 | 40.45 |

Table 10: STARK with different context ratios. Bold row is the default setting. **CR**: context ratio, **SRR**: search region ratio, **SIS**: search image size (in image resolution).

| Setting | Method | | | AO | F-score | Precision | Recall |
|---|---|---|---|---|---|---|---|
| | CR | SRR | SIS | | | | |
| Same SIS | 1x | 2.5x | 320 | 28.22 | 26.81 | 28.68 | 25.16 |
| | **2x** | **5x** | **320** | 38.94 | 33.53 | 39.13 | 29.33 |
| | 3x | 7.5x | 320 | 44.70 | 36.03 | 40.28 | 32.59 |
| | 4x | 10x | 320 | 43.19 | 34.32 | 37.98 | 31.31 |
| Same SRR | 1x | 5x | 640 | 41.50 | 31.09 | 30.31 | 31.91 |
| | 3x | 5x | 208 | 39.87 | 35.36 | 41.54 | 30.79 |
| Same CR | 2x | 7.5x | 480 | 48.21 | 39.69 | 43.95 | 36.19 |
| | 2x | 10x | 640 | 52.09 | 42.39 | 46.23 | 39.15 |
| | 2x | 12.5x | 800 | 54.08 | 43.74 | 47.60 | 40.45 |

## 4.5 Evaluating EgoSTARK on EPIC-VISOR

To demonstrate egocentric object tracking generalization from training on EgoTracks, we perform further quantitative evaluations on the EPIC-VISOR dataset [14], an egocentric dataset focused on active objects in kitchen scenes. We observe that training with EgoTracks leads to both Siam R-CNN and STARK showing improvements in tracking on EPIC-VISOR (Table 8), demonstrating the value of EgoTracks as a large-scale pre-training dataset. By additionally incorporating the improvements we propose in Section 5, EgoSTARK achieves further improvements on EPIC-VISOR.

## 5 Egocentric tracking design considerations

Observing that existing trackers do not perform well on EgoTracks, we perform a systematic exploration of priors and other design choices for egocentric tracking. Though not specifically designed for long-term tracking, Section 4 suggests STARK [84] to be the most competitive tracker on EgoTracks. We focus on this tracker for additional analysis, suggesting improvements to egocentric performance.

### 5.1 Egocentric finetuning is essential

We first demonstrate how various trackers trained on third-person videos can significantly benefit from finetuning on EgoTracks. As shown in Table 7, all methods gain improvement on F-score ranging from 6% - 10%. In addition, as shown in Table 9, finetuning on the VQ response track subset improves the F-score from 30.48% to 33.53%, while using the full EgoTracks annotation further improves the F-score by 4.67% to 38.2%. This demonstrates that: 1) finetuning with egocentric data helps close the exocentric-egocentric domain gap; 2) training on full EgoTracks provides further gains, showing the value of our training set.

### 5.2 Third-person spatiotemporal priors fail

Modern SOTs find certain assumptions on object motion, appearance, and surroundings helpful on past datasets, but some of these design choices translate poorly to long-term egocentric videos.

**Search window size.** An example is local search. Many trackers assume the tracked object appears within a certain range of its previous location. Thus, for efficiency, these methods often search within a local window of the next frame. This is reasonable in high FPS, smooth videos with relatively slow motion, commonly in previous short-term tracking data, but in egocentric videos, the object's pixel coordinates can change rapidly (frequent large head motions), and re-detection becomes a key problem. Therefore, we experiment with expanded search regions beyond what are common in past methods. As we expand search size from 320 to 800, we see dramatic improvements (Table 9): STARK is able to locate objects that were previously outside search window due to rapid movements.

**Multiscale augmentations.** The characteristics of egocentric video also affect common SOT assumptions of object scale. Many trackers are trained with the assumption that an object's scale is consistent with the template image and between adjacent frames. However, large egocentric camera

motions, locomotion, and hand interactions with objects (*e.g.* bringing an object to one's face, as in eating) can translate to objects rapidly undergoing large changes in scale. We thus propose adding scale augmentations during training, randomly resizing the search image by a factor of $s \in [0.5, 1.5]$. While simple, we find this dramatically improves performance on EgoTracks, improving STARK's AO by nearly $10\%$ and F-score by more than $8\%$ (Table 9).

**Context ratio.** Past SOT works have found that including some background can be helpful for template image feature extraction, with twice the size of the object being common. We experiment with different context ratios to see if this rule of thumb transfers to egocentric videos. Because of the local window assumption, the sizes of the template and search images are related: $\frac{\text{Search Image Size(SIS)}}{\text{Search Region Ratio(SRR)}} = \frac{\text{Template Image Size}}{\text{Context Ratio(CR)}} = \text{Object Scale}$. The template image size is set to a fixed size $128 \times 128$. When changing the context ratio, we carefully control the other parameters for a fair comparison. The results are shown in Table 10. Among all three parameters - **CR**, **SRR**, and **SIS**, the search region size (determined by **SRR** and **SIS**) has the highest impact on the F-score. This is expected because there are frequent re-detections, which require the tracker to search in a larger area for the object, rather than just within the commonly used local window. Varying the **CR** has mixed results so we adhere to the common practice of using a **CR** of 2.

# 6   Conclusion

We present EgoTracks, the first large-scale dataset for long-term egocentric visual object tracking in diverse scenes. We conduct extensive experiments to understand the performance of state-of-the-art trackers on this new dataset, and find that they struggle considerably, possibly in part due to overfitting to some of the simpler characteristics of existing benchmarks. We thus propose several adaptations for the egocentric domain, leading to a strong baseline that we call Ego-STARK, which has vastly improved performance on EgoTracks. By publicly releasing this dataset, we hope to encourage advancements in the field of long-term tracking and draw more attention to the challenges of long-term and egocentric videos.

**Challenges and future directions.** Based on our experiments in Tables 4 and 5, we find re-detection to be a key challenge of long-term tracking, especially in egocentric video, where objects frequently go in and out of view, or are exposed to high motion blur. We see a few promising future directions:

- **Stronger features**: The sizable gap in performance between the Oracle and InstEmb variants of Tracking by Detection (Table 4) illustrates the impact of insufficiently discriminative features. Better features, for example with geometric keypoints, optical flow, or long-term trajectories [77], would allow improved association of objects and thus re-detection.
- **Leveraging spatial signals**: Camera trajectories can provide powerful spatiotemporal priors to trackers. For example, knowledge of camera pose can help re-localize a object whose position does not change while out of view.
- **Global, multi-view object representations**: Egocentric videos, with their diverse camera trajectories and tendency to capture interactions with objects, often offer significantly more varied viewpoints of objects than traditional third-person tracking datasets. In the latter, object appearances tend to be more constant, so prior tracking methods can often get away with a single image template. The aforementioned charactistics of egocentric video necessitate more robustness, and a challenging egocentric tracking dataset like EgoTracks represents an opportunity to develop more global, view-variant object representations learned in an online fashion. A simple version can be found in Appendix D, where we augmented EgoSTARK by fusing multiple templates, which we find indeed improved EgoTracks results.

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

# A  EgoTracks data details

## A.1  Data splits

We followed the same data split as [29] for train, val and test sets. The exact number for the number of videos and number of tracks may not be the same as [29], because annotators were instructed to ignore videos and objects that they may not know how to annotate due to reasons like poor video resolution, failure to understand what object to track and so on.

Table 11: Data split for EgoTracks. We follow the same split as Ego4D VQ benchmark[29].

| Split | # of videos | # of tracks |
|-------|-------------|-------------|
| Train | 3433 | 13140 |
| Val | 1154 | 4459 |
| Test | 1121 | 4429 |
| Total | 5708 | 22028 |

## A.2  Object size distribution

Figure 6 shows the distribution of object sizes in EgoTracks. To account for the difference in video resolution in Ego4D, we use normalized object size instead of absolute pixel size to plot the distribution. The normalized object size is defined as the object size divided by image resolution: $\sqrt{obj_w \times obj_h}/\sqrt{img_w \times img_h}$.

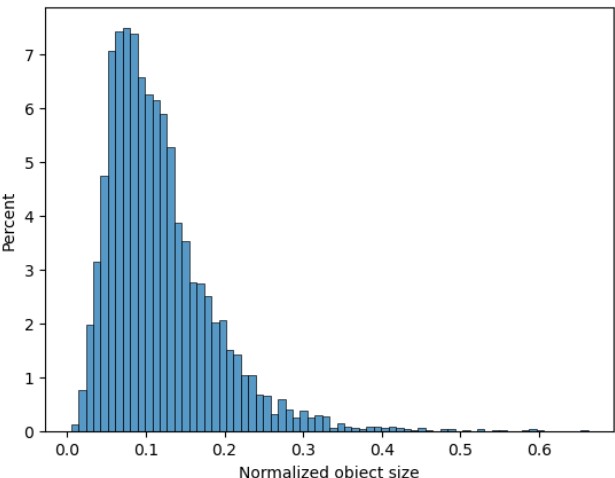

Figure 6: Object size distribution of EgoTracks. The object size is normalized by video resolution to account for differences in resolution in Ego4D.

# B  Tracking by detection

In Section 4.2 of the main paper, we introduced a few baselines leveraging the paradigm of Tracking-by-Detection. We detail our designs and implementations in this section.

A Tracking-by-Detection framework contains two modules:

- A **Detector** that generates bounding box proposals. Each proposed bounding box contains an instance that potentially corresponds to the tracking target. Since SOT focuses on generic objects, this detector module should be as generic as possible, not limited to specific types of objects.
- An **Association** module that links proposed bounding boxes to template. This module takes proposals from the detector and chooses the one that most likely corresponds to the template object.

### B.1 Detector models

We benchmark two off-the-shelf models: a baseline Mask R-CNN [30] and an open-world instance segmentation model GGN [78]. Both models are trained on COCO2017 [49] dataset with 80 object categories. The models are trained in a class-agnostic fashion (object vs. background) since it generalizes better on generic objects [79]. We choose an open-world model since we would like a detector that can detect objects regardless of their categories, one of the main advantage of open-world detectors. GGN uses a similar architecture as Mask R-CNN but achieves SOTA on multiple benchmarks through new training strategies dedicated to open-world. We use the bounding boxes from the box head as bounding box proposals. To evaluate detectors, we use an Oracle to select the best box matched to groundtruth (by maximum overlap with groundtruth).

Table 12: **Comparing detectors on EgoTracks.** Open-world detector GGN strongly outperforms baseline Mask R-CNN. @k means we select top-k detecting results.

| Method | AO@10 | AO@30 | AO@100 |
|---|---|---|---|
| Mask R-CNN | 24.27 | 42.76 | 60.00 |
| GGN | **47.59** | **64.01** | **75.92** |

Both model achieve reasonable performance (Table 12). The AO@100 is much higher than tracking baselines. The open-world GGN model achieves much stronger performance on both low number of proposals (AO@10) and high number of proposals (AO@100). This shows that the ability to detect generic objects regardless of their categories is indeed an important attribute of EgoTracks. In addition, this also proves that EgoTracks contains many objects out of the popular COCO taxonomy, as Mask R-CNN performs significantly worse.

### B.2 Associating bounding boxes

We use bounding box (instance) embedding to select from proposed bounding boxes as tracking output. Specifically, we compute embedding features on the cropped images from proposed bounding boxes and the template image. Then we compute cosine similarities between the template feature and all bounding box features. We select the bounding box with highest cosine similarity as tracking output, and the cosine similarity is output as tracking confidence.

We consider two models to extract embedding. **(1)** We use a standard ResNet-50 trained on ImageNet as a **baseline**. **(2)** To enforce discrimination across different instances, we use a ResNet-50 model trained with additional instance discrimination task: **InstEmb**. On top of a standard cross-entropy loss similar as the baseline model, we add a supervised contrastive loss [37] by contrasting across instances (both same categories and different categories). To generate more instance examples, we add Objects365 [69] to training data by cropping images based on bounding box annotations.

Table 13: **Performance of different embedding models.** We observe InstEmb model performs significantly better, but still has a large gap with the oracle in Table 12 on AO. This shows that temporal association is very challenging in EgoTracks.

| Method | AO | F1 | Precision | Recall |
|---|---|---|---|---|
| Baseline | 2.5 | 1.2 | 0.8 | 2.5 |
| InstEmb | **15.2** | **9.9** | **11.8** | **8.6** |

We use the top-100 bounding boxes from GGN model. We summarize the two models in Table 13. We observe that using the baseline ImageNet features do not generate good association between detection results and template object. The InstEmb model achieves much stronger performance, showing the importance of instance discrimination from the additional instance contrastive loss. Despite its stronger performance, there is a significant gap compared to the oracle and most tracking methods. This shows that the temporal association problem is very challenging in EgoTracks.

## C   Effect of visual template for short-term tracking

We study the effect of different STARK template initializations to demonstrate how the template affects the tracking. More specifically, we experiment with two protocols: **VCT-$\overrightarrow{OO}$** and **OFFT-$\overrightarrow{OO}$**.

Table 14: Comparing initializing STARK with different templates.

| Method | AO | Success | Pre | $Pre_{norm}$ |
|---|---|---|---|---|
| STARK - **OFFT-$\overrightarrow{OO}$** | 48.18 | 48.86 | 45.76 | 51.90 |
| STARK - **VCT-$\overrightarrow{OO}$** | 51.10 | 51.83 | 47.78 | 53.96 |

Table 15: STARK with different numbers of templates.

| Templates | AO | F-score | Precision | Recall |
|---|---|---|---|---|
| 1 | 32.97 | 25.42 | 25.80 | 25.04 |
| 3 | 34.76 | 26.84 | 28.84 | 25.57 |
| 5 | 35.47 | 28.03 | 29.82 | 26.45 |
| 7 | 34.81 | 27.83 | 30.77 | 25.40 |
| 9 | 33.92 | 26.89 | 30.36 | 24.12 |

The difference between the two protocols is the template used for initializing the tracker: **VCT-$\overrightarrow{OO}$** uses the visual crop, while **OFFT-$\overrightarrow{OO}$** uses the first frame for each occurrence. To ensure a fair comparison between the two protocols, we ignore both the visual crop and the first frame for each occurrence when evaluating. Results are shown in Table 14. Please note the STARK - **OFFT-$\overrightarrow{OO}$** in Table 14 uses the same model as STARK in Table 5. The difference in performance is due to ignoring the visual crop in evaluation in Table 14. Even though the visual crop is not adjacent temporally to the start of each occurrence, **VCT-$\overrightarrow{OO}$** performs 3% better to **OFFT-$\overrightarrow{OO}$** in terms of AO because of the better template used.

# D  Multiple templates can improve tracking

Transformer-based architectures can encode arbitrary length inputs, making it straightforward to consume features from an arbitrary number of templates. The original STARK design encodes two templates: the initialization and a single dynamically updated template. A natural extension is to include more templates, which may expose the transformer to different views of the object (particularly relevant in egocentric video), though low-quality views may compromise performance [47].

What's the right trade-off? We experiment with different numbers of templates for a basic STARK model. Motivated by potential applications where a user can take a short video of an object from different angles [65], we extend the single visual crop to a visual clip of templates by incorporating additional frames from the same occurrence where the visual crop appears as the template. We adopt a simple template sampling method: uniformly sampling 3, 5, 7, or 9 templates from the visual crop's occurrence. Uniformly sampling the videos temporally can be a simple yet effective heuristic to gather diverse views from an occurrence. We summarize the results in Table 15. While we observe improvements across all metrics using up to 5 templates, performance declines with more. We hypothesize that increasing the number of templates does increase the knowledge available to STARK for tracking, but after a certain point it may dilute the information in the templates and make it difficult for the transformer to synthesize. This highlights the importance of template selection and multi-view fusion mechanisms, which inspires promising directions.

# E  Re-detection success rate

To more directly capture re-detection rate of trackers on EgoTracks, we pose it as a per-occurrence detection problem, where we consider that the tracker has successfully re-detected an object during an occurrence if there is at least one frame during the occurrence where the tracker produced a bounding box above some IOU threshold with the ground truth. We report the average proportion of occurrences for which the tracker meets this criterion below, at two IOU thresholds: 0.5 and 0.75. Table 16 confirms that recall (F-score) is consistent with the success rate of re-detection: methods with higher recall (therefore F-Score) also have higher re-detection success rate.

Table 16: Success rate of re-detection rate for different methods on EgoTracks.

| Method | Recall@IoU 0.5 | Recall@IoU 0.75 |
|---|---|---|
| KYS | 21.90 | 16.54 |
| DiMP | 22.35 | 16.77 |
| GlobalTrack | 42.72 | 35.55 |
| LTMU | 41.18 | 34.67 |
| ToMP | 39.85 | 34.81 |
| Siam-RCNN | 47.73 | 44.70 |
| MixFormer | 37.57 | 25.61 |
| STARK - Res50 | 43.31 | 39.00 |
| STARK - Res101 | 42.40 | 38.34 |
| EgoSTARK | 66.83 | 59.95 |

## F  Additional visualizations

We provide additional visualizations for four tracking methods introduced in Section 4.2 for the entire video sequence, as well as the ground truth annotation: GlobalTrack [34], LTMU [11], ToMP [55], STARK [84]. Due to size limit, we chose four example videos with hand and object interaction.

## G  Resources

**Computation.** For finetuning STARK, we used one machine with 8 V100 GPUs, with a running time of roughly one day. For all other baselines, we only run inference, using 4 machines each with 8 V100 GPUs.

**Annotation.** Each video has an average of 3.8 objects, and each object track took between 1-2 hours to annotate the bounding boxes. The attributes annotation took around 30 seconds per object track. We estimate the total number of hours of annotation to be around 86k, including human annotations and quality check.

## H  Limitations

The bounding box annotation was done by human labelers. Although multiple rounds of reviews were used to ensure correctness and consistency, there still remains the potential for human errors or bias. In addition, the selection of objects is inherited from the Ego4D visual-query benchmark, which is biased towards objects with appearances of at least modest duration. Consequently, our benchmark may contain less instances that only appear very briefly in the video. However, the dataset does contain a moderate amount of occurrences for objects with short duration.

## I  Potential negative societal impacts

Since our videos are sourced from Ego4D, we inherit many of the potential negative impacts. Details are discussed in the original paper [29] Appendix K. We summarize several here:

- There may be risks surrounding privacy, such as personnel being recorded during the video. Consent was obtained in the original dataset, and a user agreement is enforced for Ego4D. Our dataset follow the same protocol as Ego4D.

- The existing efforts may inspire future data collection with less attention to privacy and ethics. Best practices were detailed in the original paper [29]. We did the same with our paper to include the instructions to help mitigate this risk.

- There may be data imbalances, such as geographical distribution. This risk can be mitigated with future work that grows the collaboration in underrepresented areas.

# J  Data sheet

We follow [25] for writing the data sheet of EgoTracks.

1. Motivation

    (a) *For what purpose was the dataset created? (Was there a specific task in mind? Was there a specific gap that needed to be filled? Please provide a description.)*
    Visual object tracking is key to many egocentric vision problems. However, the full spectrum of challenges of egocentric tracking faced by an embodied AI is underrepresented in many existing datasets, which tend to focus on short, third-person videos. Egocentric video has several distinguishing characteristics from those commonly found in past datasets: frequent large camera motions and hand interactions with objects commonly lead to occlusions or objects exiting the frame, and object appearance can change rapidly due to widely different points of view, scale, or object states. Embodied tracking is also naturally long-term, and being able to consistently (re-)associate objects to their appearances and disappearances over as long as a lifetime is critical. Previous datasets under-emphasize this re-detection problem, and their "framed" nature has led to adoption of various spatiotemporal priors that we find do not necessarily generalize to egocentric video. We thus introduce EgoTracks, a new dataset for long-term egocentric visual object tracking. Sourced from the Ego4D dataset, EgoTracks presents a significant challenge to recent state-of-the-art single-object trackers, which we find score more poorly on our new dataset than existing popular benchmarks, according to traditional tracking metrics.

    (b) *Who created this dataset (e.g., which team, research group) and on behalf of which entity (e.g., company, institution, organization)?*
    EgoTracks was created by Hao Tang, Kevin J Liang, Kristen Grauman, Matt Feiszli, and Weiyao Wang. All the authors were employed by Meta Platforms Inc. Kristen Grauman is also a professor at UT Austin.

    (c) *Who funded the creation of the dataset? (If there is an associated grant, please provide the name of the grantor and the grant name and number.)*
    Funding was provided by Meta Platforms Inc.

    (d) *Any other comments?*
    The code and dataset annotation is available for download from https://github.com/EGO4D/episodic-memory/tree/main/EgoTracks.

2. Composition

    (a) *What do the instances that comprise the dataset represent (e.g., documents, photos, people, countries)? (Are there multiple types of instances (e.g., movies, users, and ratings; people and interactions between them; nodes and edges)? Please provide a description.)*
    Each instance is an object track from a video from the Ego4D dataset [29]. Object tracks are each represented as a series of bounding boxes across multiple frames.

    (b) *How many instances are there in total (of each type, if appropriate)?*
    There are 22028 instances in total. Please see Table 11 for details on how we split the dataset for training, validation, and test.

    (c) *Does the dataset contain all possible instances or is it a sample (not necessarily random) of instances from a larger set? (If the dataset is a sample, then what is the larger set? Is the sample representative of the larger set (e.g., geographic coverage)? If so, please describe how this representativeness was validated/verified. If it is not representative of the larger set, please describe why not (e.g., to cover a more diverse range of instances, because instances were withheld or unavailable).)*
    The dataset does not contain all possible instances in the videos. Instead, objects were selected according to those present in Ego4D's Visual Query benchmark, where object instances were chosen by annotators if they think the object is "interesting", which means the object that they think they might interact with or is important for an activity (see [29] for details). Since these object instances were inherited from Ego4D, we share the same human and geographic bias and limitations in object selection (see Appendix H).

(d) *What data does each instance consist of? ("Raw" data (e.g., unprocessed text or images) or features? In either case, please provide a description.)*
Each instance consists of 2D bounding boxes $(x, y, w, d)$ for the object for each frame in the video where the object is visible. Also, for each time interval the object appears in the video, there are three binary labels: `is_active`, `is_transformed`, and `is_recognizable` (see Section 3.3 of main paper for definitions).

(e) *Is there a label or target associated with each instance? If so, please provide a description.*
For each time interval the object appears in the video, there are three binary labels: is_active, is_transformed and is_recognizable. is_active means if the camera wearer is interacting with the object. is_transformed means if the object is undergoing any transformation (rigid or non-rigid) or state change. is_recognizable means if the annotator can recognize the object given the time interval. Otherwise, the location specified by the bounding boxes comprises the label.

(f) *Is any information missing from individual instances? (If so, please provide a description, explaining why this information is missing (e.g., because it was unavailable). This does not include intentionally removed information, but might include, e.g., redacted text.)*
No, we do not remove any information for individual instances already present in Ego4D [29].

(g) *Are relationships between individual instances made explicit (e.g., users' movie ratings, social network links)? ( If so, please describe how these relationships are made explicit.)*
No. There are on average roughly 3.86 tracks per track, and groups of videos in Ego4D [29] may have been captured by the same individual, but we do not expose this information in the dataset.

(h) *Are there recommended data splits (e.g., training, development/validation, testing)? (If so, please provide a description of these splits, explaining the rationale behind them.)*
Yes, we follow the same data splits as [29]. Please refer to Table 11 for details.

(i) *Are there any errors, sources of noise, or redundancies in the dataset? (If so, please provide a description.)*
Since the bounding boxes are annotated by human raters, they are prone to human bias and errors. We applied quality assurance procedures to minimize such errors where we can.

(j) *Is the dataset self-contained, or does it link to or otherwise rely on external resources (e.g., websites, tweets, other datasets)? (If it links to or relies on external resources, a) are there guarantees that they will exist, and remain constant, over time; b) are there official archival versions of the complete dataset (i.e., including the external resources as they existed at the time the dataset was created); c) are there any restrictions (e.g., licenses, fees) associated with any of the external resources that might apply to a future user? Please provide descriptions of all external resources and any restrictions associated with them, as well as links or other access points, as appropriate.)*
The underlying videos are sourced from [29]. The [29] dataset is maintained by Ego4D consortium which can be regarded as guaranteed to exist and remain constant. The license can be found https://ego4d-data.org/pdfs/Ego4D-Licenses-Draft.pdf.

(k) *Does the dataset contain data that might be considered confidential (e.g., data that is protected by legal privilege or by doctor-patient confidentiality, data that includes the content of individuals' non-public communications)? (If so, please provide a description.)*
No. Ego4D [29] was collected with careful consideration of ethics and consent. We largely inherit these characteristics. The additional bounding box and attribute annotations of our dataset do not reveal any confidential information.

(l) *Does the dataset contain data that, if viewed directly, might be offensive, insulting, threatening, or might otherwise cause anxiety? (If so, please describe why.)*
We share the same underlying videos as [29], so we inherit its privacy and ethics standards, as well as any potential risks. The additional bounding box and attribute annotations of our dataset do not add any additional objectionable content.

(m) *Does the dataset relate to people? (If not, you may skip the remaining questions in this section.)*

Yes. The annotations we provide in EgoTracks do not contain any information relating to people; however, they are labeled from Ego4D [29] videos, which do contain people, and indirectly are related to people due to its egocentric nature.

(n) *Does the dataset identify any subpopulations (e.g., by age, gender)? (If so, please describe how these subpopulations are identified and provide a description of their respective distributions within the dataset.)*

EgoTracks does not identify subpopulations, but the underlying dataset Ego4D [29] does. See Appendix C of [29] for more details.

(o) *Is it possible to identify individuals (i.e., one or more natural persons), either directly or indirectly (i.e., in combination with other data) from the dataset? (If so, please describe how.)*

EgoTracks does not contain tracks of people, focusing instead on objects. Ego4D [29] does contain some visually identifiable individuals, who provided their consent to appear in the dataset. Other individuals have their faces blurred.

(p) *Does the dataset contain data that might be considered sensitive in any way (e.g., data that reveals racial or ethnic origins, sexual orientations, religious beliefs, political opinions or union memberships, or locations; financial or health data; biometric or genetic data; forms of government identification, such as social security numbers; criminal history)? (If so, please provide a description.)*

EgoTracks does not add any labels for people, focusing instead on objects. Ego4D [29] does contain demographics information. See (n) above.

(q) *Any other comments?*

N/A.

3. Collection Process

(a) *How was the data associated with each instance acquired? (Was the data directly observable (e.g., raw text, movie ratings), reported by subjects (e.g., survey responses), or indirectly inferred/derived from other data (e.g., part-of-speech tags, model-based guesses for age or language)? If data was reported by subjects or indirectly inferred/derived from other data, was the data validated/verified? If so, please describe how.)*

The bounding box for each instance was acquired by human annotators. They were instructed to draw a 2D bounding box for each frame the object appears in the video. To ensure data correctness and consistency, three independent reviewers are asked to annotate the same video. We find the overlaps between these independent annotations are high ($>0.88$ IoU).

(b) *What mechanisms or procedures were used to collect the data (e.g., hardware apparatus or sensor, manual human curation, software program, software API)? (How were these mechanisms or procedures validated?)*

We use the proprietary annotation software from Meta to collect bounding box annotations. The software shows a video frame by frame, and the annotator is able to use mouse to draw a bounding box around the object. The software will then record the coordinate for each bounding box.

(c) *If the dataset is a sample from a larger set, what was the sampling strategy (e.g., deterministic, probabilistic with specific sampling probabilities)?*

We used the entirety of Ego4D's Visual Queries benchmark [29] as the basis of Ego-Tracks.

(d) *Who was involved in the data collection process (e.g., students, crowdworkers, contractors) and how were they compensated (e.g., how much were crowdworkers paid)?*

The participants were contractors employed by a third-party vendor and are compensated based on the agreement with their employer.

(e) *Over what timeframe was the data collected? (Does this timeframe match the creation timeframe of the data associated with the instances (e.g., recent crawl of old news articles)? If not, please describe the timeframe in which the data associated with the instances was created.)*

The dataset was created in summer and fall of 2022, which is not the time the videos were collected.

(f) *Were any ethical review processes conducted (e.g., by an institutional review board)? (If so, please provide a description of these review processes, including the outcomes, as well as a link or other access point to any supporting documentation.)*

Yes. For proprietary reasons, we are not able to provide supporting documentation. Our internal review was conducted thoroughly vetted potential privacy and ethical related concerns.

(g) *Does the dataset relate to people? (If not, you may skip the remaining questions in this section.)*

Yes. The annotations we provide in EgoTracks do not contain any information relating to people; however, they are labeled from Ego4D [29] videos, which do contain people, and indirectly are related to people due to its egocentric nature.

(h) *Did you collect the data from the individuals in question directly, or obtain it via third parties or other sources (e.g., websites)?*

Ego4D [29] collected the data from the individual directly. We do not collect any additional videos beyond those in Ego4D.

(i) *Were the individuals in question notified about the data collection? (If so, please describe (or show with screenshots or other information) how notice was provided, and provide a link or other access point to, or otherwise reproduce, the exact language of the notification itself.)*

Ego4D [29] was collected by willing and consenting individuals. See Section 3.4 in [29].

(j) *Did the individuals in question consent to the collection and use of their data? (If so, please describe (or show with screenshots or other information) how consent was requested and provided, and provide a link or other access point to, or otherwise reproduce, the exact language to which the individuals consented.)*

Ego4D [29] was collected by willing and consenting individuals. See Section 3.4 in [29].

(k) *If consent was obtained, were the consenting individuals provided with a mechanism to revoke their consent in the future or for certain uses? (If so, please provide a description, as well as a link or other access point to the mechanism (if appropriate).)*

Yes, see Section 3.4 in [29].

(l) *Has an analysis of the potential impact of the dataset and its use on data subjects (e.g., a data protection impact analysis) been conducted? (If so, please provide a description of this analysis, including the outcomes, as well as a link or other access point to any supporting documentation.)*

No.

(m) *Any other comments?*

None.

4. Preprocessing/cleaning/labeling

(a) *Was any preprocessing/cleaning/labeling of the data done (e.g., discretization or bucketing, tokenization, part-of-speech tagging, SIFT feature extraction, removal of instances, processing of missing values)? (If so, please provide a description. If not, you may skip the remainder of the questions in this section.)*

No.

(b) *Was the "raw" data saved in addition to the preprocessed/cleaned/labeled data (e.g., to support unanticipated future uses)? (If so, please provide a link or other access point to the "raw" data.)*

No.

(c) *Is the software used to preprocess/clean/label the instances available? (If so, please provide a link or other access point.)*

No.

(d) *Any other comments?*

None.

5. Uses

   (a) *Has the dataset been used for any tasks already? (If so, please provide a description.)*
   The dataset has been used for understanding the performance of different single object tracking (SOT) trackers, and developing trackers that perform better in egocentric settings.

   (b) *Is there a repository that links to any or all papers or systems that use the dataset? (If so, please provide a link or other access point.)*
   We released our code for EgoSTARK, along with the dataset in `https://github.com/EGO4D/episodic-memory/tree/main/EgoTracks`.

   (c) *What (other) tasks could the dataset be used for?*
   The dataset could possibly be used for acquiring segmentation masks, constructing 3D representation for the objects, or any other tasks that can use object tracks.

   (d) *Is there anything about the composition of the dataset or the way it was collected and preprocessed/cleaned/labeled that might impact future uses? (For example, is there anything that a future user might need to know to avoid uses that could result in unfair treatment of individuals or groups (e.g., stereotyping, quality of service issues) or other undesirable harms (e.g., financial harms, legal risks) If so, please provide a description. Is there anything a future user could do to mitigate these undesirable harms?)*
   The selection of objects is inherited from the Ego4D visual-query benchmark, which is biased towards objects with appearances of at least modest duration. Consequently, our benchmark may contain less instances that only appear very briefly in the video. However, the dataset does contain a moderate amount of occurrences for objects with short duration.

   (e) *Are there tasks for which the dataset should not be used? (If so, please provide a description.)*
   We currently do not foresee anything such tasks.

   (f) *Any other comments?*
   None.

6. Distribution

   (a) *Will the dataset be distributed to third parties outside of the entity (e.g., company, institution, organization) on behalf of which the dataset was created? (If so, please provide a description.)*
   Yes, we share the same license and distribution as [29].

   (b) *How will the dataset will be distributed (e.g., tarball on website, API, GitHub)? (Does the dataset have a digital object identifier (DOI)?)*
   The dataset is distributed from GitHub `https://github.com/EGO4D/episodic-memory/tree/main/EgoTracks`.

   (c) *When will the dataset be distributed?*
   It is currently available on GitHub.

   (d) *Will the dataset be distributed under a copyright or other intellectual property (IP) license, and/or under applicable terms of use (ToU)? (If so, please describe this license and/or ToU, and provide a link or other access point to, or otherwise reproduce, any relevant licensing terms or ToU, as well as any fees associated with these restrictions.)*
   We share the same license as Ego4D. Please see `https://ego4d-data.org/pdfs/Ego4D-Licenses-Draft.pdf`.

   (e) *Have any third parties imposed IP-based or other restrictions on the data associated with the instances? (If so, please describe these restrictions, and provide a link or other access point to, or otherwise reproduce, any relevant licensing terms, as well as any fees associated with these restrictions.)*
   Not to our knowledge.

   (f) *Do any export controls or other regulatory restrictions apply to the dataset or to individual instances? (If so, please describe these restrictions, and provide a link or other access point to, or otherwise reproduce, any supporting documentation.)*
   Not to our knowledge.

   (g) *Any other comments?*
   None.

7. Maintenance

   (a) *Who is supporting/hosting/maintaining the dataset?*
      All authors are maintaining the dataset. The Ego4D team is also supporting the dataset.

   (b) *How can the owner/curator/manager of the dataset be contacted (e.g., email address)?*
      Authors can be contacted via emails, or via the Ego4D forum `https://discuss.ego4d-data.org/top?period=yearly`.

   (c) *Is there an erratum? (If so, please provide a link or other access point.)*
      Not currently. Future versions of the dataset may be released if we find errors, which will be provided within the same GitHub.

   (d) *Will the dataset be updated (e.g., to correct labeling errors, add new instances, delete instances')? (If so, please describe how often, by whom, and how updates will be communicated to users (e.g., mailing list, GitHub)?)*
      See previous question.

   (e) *If the dataset relates to people, are there applicable limits on the retention of the data associated with the instances (e.g., were individuals in question told that their data would be retained for a fixed period of time and then deleted)? (If so, please describe these limits and explain how they will be enforced.)*
      N/A. The dataset is not related to people.

   (f) *Will older versions of the dataset continue to be supported/hosted/maintained? (If so, please describe how. If not, please describe how its obsolescence will be communicated to users.)*
      Yes, all data will be versioned.

   (g) *If others want to extend/augment/build on/contribute to the dataset, is there a mechanism for them to do so? (If so, please provide a description. Will these contributions be validated/verified? If so, please describe how. If not, why not? Is there a process for communicating/distributing these contributions to other users? If so, please provide a description.)*
      Errors/features can be submitted as issues/pull requests on GitHub.

   (h) *Any other comments?*
      None.

