# OpenReview forum: "EgoTracks: A Long-term Egocentric Visual Object Tracking Dataset"
_NeurIPS.cc/2023/Track/Datasets_and_Benchmarks — NeurIPS 2023 Datasets and Benchmarks Poster_

### Official Review · Reviewer_G7Uz · 2023-07-06

**Rating:** 6
**Confidence:** 4

**Strengths:**

The authors propose first-person view datasets sourced from Ego4D with the annotations. There are many VOT datasets in public with third-person view not the first-person view. Also, even there are some long-term single object tracking datasets which has large occlusions and disappearance&re-appearance, the scale is significantly large with long video sequences with more challenging scenarios by more disappearances & reappearances and diverse perspective changes.
Also, they offers the extensive experiments results to understand the performance of the current deep trackers on this new dataset such as MixFormer, STARK, and MaskR-CNN. And it results to the proposed datasets is very challenging to the current trackers which is one of the strengths of the proposed work.


**Additional Feedback:**

I wonder the performance of the Ego-STARK on the other tracking benchmark. Please, refers the opportunities for improvement part.

**Clarity:**

I think this study is well written in general, and some parts need to be modified.

**Correctness:**


They offers the various evaluation protocol including different choice of tracker initialization, and temporal direction. If they offers the compare of the robustness/other nature of the metric VCT-VC, VCT-VS, and the others, it would be more helpful to the reader to choose which protocol to evaluate their tracker.


**Documentation:**

The documentation is well written to support reproduciblity sufficiently.

**Ethics:**

I have no suspect on their ethical concerns with their submission.

**Limitations:**

Please, refers the Opportunities for improvement part.

**Opportunities For Improvement:**

As the author mentioned, the proposed dataset consists of many long-term sequences with a lot of occlusion, disappearances/re-appearances with large motion on the frame. Therefore, the performance of re-detection is the most important aspect of the object tracker to get the high score in this dataset. However, many trackers without re-detection algorithms are evaluated and more tracking by detection algorithms should be evaluated on the proposed datasets.

They find that the current deep trackers struggle considerably, possibly in part due to over-fitting to some of the simpler characteristics of existing benchmarks. And they suggest Ego-STARK to proposed their effectiveness of the dataset. I wonder Ego-STARK shows better performance on the other tracking benchmarks including long-term and short-term datasets. If Ego-STARK solve the problem of the over-fitting to the simpler characteristics of certain dataset, it produces same or superior performances on the other datasets.

**Relation To Prior Work:**

The authors mention the differences between the egocentric videos and the general datasets for the object tracking[21,32,57,75], because the large motions from the camera often result in objects repeatedly leaving and re-entering the field of view. And the proposed datasets is a annotated subset of Ego4D and compared to the [12], it has larger-scale and long-term videos. The authors introduce EgoTracks in detail to clearly discuss how the proposed datasets are constructed from the dataset.

**Summary And Contributions:**

They proposed dataset for long-term egocentric visual object tracking called EgoTracks, which emphasize the re-detection problem caused by the disappearance and re-appearance of the target object in the frames. Their nature comes from usually large camera motions and hand interactions with object commonly lead to occlusions/out of view of the target object. And also, they propose the modified version of STARK for the egocentric data, EgoSTARK to deal with these properties of ego-centric videos.

---

> ### Author Response · Authors · 2023-08-16
>
> 1. Trackers with re-detection: We emphasize that several baselines in our paper (GlobalTrack, LTMU, and SiamRCNN) are in fact recent SoTA long-term trackers with re-detection capabilities; the Tracking by Detection approaches in the bottom half of Table 4 can also be considered as having re-detection capabilities. We’ve made changes to Table 4 in our revision to make this distinction clearer. As the reviewer points out, re-detection is an important component of EgoTrack’s challenging nature, though we do also want to emphasize it’s not the only consideration either. For example, in the $OFFT-\overrightarrow{OO}$ evaluation protocol, we remove the need for re-detection entirely, and yet there is still significant room for improvement from the baselines (Table 5, left).
>
> 2. EgoSTARK on other datasets: We thank the reviewer for suggesting this additional set of experiments. We further test the EgoSTARK model on EPIC-VISOR, an ego-centric dataset focused on active objects in kitchen scenarios. We noticed that by training with EgoTracks, both Siam R-CNN and STARK improve on the EPIC-VISOR dataset. This demonstrates the value of EgoTracks for large-scale pre-training. In addition, by incorporating the improvements we made to STARK, EgoSTARK achieves significant improvements on EPIC-VISOR. This experiment is also added to Supplementary Material Section F.
>
>     | Method                             | AO   | F1   | Precision | Recall |
>     |------------------------------------|------|------|-----------|--------|
>     | Siam-RCNN                          | 51.0 | 42.4 | 37.7      | 48.4   |
>     | Siam-RCNN (finetuned on EgoTracks) | 52.7 | 43.5 | 38.4      | 50.3   |
>     | STARK                              | 59.2 | 46.4 | 39.4      | 56.6   |
>     | STARK (finetuned on EgoTracks)     | 61.4 | 48.5 | 41.0      | 59.4   |
>     | EgoSTARK                           | 64.7 | 50.6 | 42.6      | 62.3   |
>
> 3. Recommendation of evaluation protocol: If we had to pick one, we would recommend the $VCT-\overrightarrow{VS}$ protocol, as it balances a high quality initial template with the challenges of tracking an object through a long sequences with many disappearances and reappearances. Indeed, our primary evaluation in Section 4.2 is under this protocol. However, we do encourage further users to utilize the other evaluation protocols as well, as they capture different aspects of a tracker’s performance; for example, $OFFT-\overrightarrow{OO}$ disentangles a tracker’s ability to track an object throughout an occurrence from the tracker’s ability to re-detect an object after it disappears.

---

### Official Review · Reviewer_Pzo5 · 2023-07-18
**review with rating-6 - upgraded to 7**

**Rating:** 7
**Confidence:** 4
**Correctness:** The paper is written well.
**Clarity:** The paper is written well.

**Strengths:**

- The new dataset is a valuable asset to the community.
- Experiments are comprehensive.
- The paper is well-written.

**Additional Feedback:**

Please refer to above boxes to improve the paper.

**Documentation:**

Documentation is good with video demo, data sheet, and more results.

**Ethics:**

Supplemental material discusses potential ethic issues.

**Limitations:**

Supplemental material discusses limitations which make sense.

**Opportunities For Improvement:**

- It is unclear what fertile directions authors suggest given the current analysis. Authors are encouraged to have a discussion.
- Figure&table captions can be refined. For example, in Figure 4, the caption should include meaningful conclusions as it is hard for readers to understand the visual results -- which trackers perform well? Why?
- The headline of Section 5.2 is misleading. While it states "third-person spatiotemporal priors fail", this subsection still concludes that using such priors improve the ego-tracking performance.

**Relation To Prior Work:**

The paper discusses relevant prior work in a good way.

**Summary And Contributions:**

The paper introduces EgoTracks, a Long-term Egocentric Visual Object Tracking Dataset. It derives from an existing dataset called Ego4D by annotating more objects throughout long videos. It analyzes representative tracking methods with multiple setups (without training on the proposed dataset). It also focuses on a state-of-the-art tracker STARK, training it on the EgoTrack training set and evaluating on the validation set. It draws conclusions showing the unique challenges related to object tracking in long egocentric videos. The paper is well-written, experiments tend to be comprehensive, setups make sense, etc.

---

> ### Author Response · Authors · 2023-08-16
>
> 1. Future Directions: We thank the reviewer for suggesting a deeper discussion of fertile directions for research. We list some thoughts below; if reviewers find them indeed valuable, we will add this to the paper:
>
>     Based on our experiments in Tables 4 and 5, we found that re-detection to be a key challenge of long-term tracking, especially in egocentric video, where objects frequently go in and out of view, or are exposed to high motion blur. We see a few promising directions for future works:
>
>     a) Stronger features for associating objects should significantly improve re-detection; the impact of insufficiently discriminative feature embeddings can be clearly seen in the major gap in Tracking by Detection performance between the Oracle and InstEmb methods at the bottom of Table 4. Geometric keypoints, optical flow, or long-term trajectories [a] can also lead to large improvements here.
>
>     b) Leveraging spatial signals: camera trajectories can be estimated as additional signals to the tracker. For example, if an object remains static during the window where it is out-of-view, knowledge of camera location can help re-localize the position of this object.
>
>     c) Global, multi-view object representations: Egocentric videos, with their diverse camera trajectories and tendency to capture the camera wearer’s interactions with objects, often offer significantly richer and more varied viewpoints of objects than traditional third-person tracking datasets. In the latter, object appearances tend to be more constant, so modern tracking methods have thus far been able to get away with using a single image template (optionally with an additional template from the latest frame). With a need for more robustness to the different viewpoints and occlusions offered by egocentric video, we believe that a challenging egocentric tracking dataset like EgoTracks represents an opportunity to develop trackers with more global, view-variant object representations learned in an online fashion. A simple version of this can be found in Section D of the supplementary material, where we augmented EgoSTARK by fusing multiple templates; we found that such a strategy indeed improved tracking results on EgoTracks.
>
> 2. Figure and Table captions: We will update these to make them clearer. The purpose of Figure 4 was to show how challenging object tracking is in egocentric views. Our goal was to show that all the trackers we tested struggle with these scenes, not that any particular method is superior to another. We have added the following caption in the revision to better explain this figure:
>
>     “Figure 4: Qualitative results of different trackers. EgoTracks presents significant challenges for all trackers, due to drastic viewpoint changes, occlusions, changes in scale, head motion etc.”
>
> 3. “Third-person spatiotemporal priors fail”: To clarify, priors are indeed helpful for tracking, but we empirically demonstrate that some priors commonly adopted by prior work overfit to the simpler scenarios common to previous third-person datasets. We find many of these do not generalize well to egocentric video and thus do not adopt the same assumptions.
>
>
> [a] Wang, Qianqian, et al. "Tracking Everything Everywhere All at Once." arXiv preprint arXiv:2306.05422 (2023).

---

> > ### Comment · Reviewer_Pzo5 · 2023-08-29
> >
> > Thanks for the responses, which largely address my concerns. As for in-depth discussions about future work, I suggest authors include (some of) them in the camera-ready, which allows 10 pages (i.e., more spaces for the discussion). Such a discussion helps others follow this dataset and develop new methods.

---

### Official Review · Reviewer_hGyN · 2023-07-21
**Initial review**

**Rating:** 8
**Confidence:** 4
**Correctness:** The dataset is constructed in a prope…
**Clarity:** Yes, it is well written.

**Strengths:**

1) The proposed dataset addresses an interesting and also challenging vision task of egocentric object tracking. The extensive experiements sufficiently demonstrate the challenges of egocentric visual object tracking and provide hints for egocentric tracking design.
2) The paper is very well written. The comparison with previous work and the motivation of the proposed dataset are clearly stated. The analysis of the experiments and the considerations of egocentric tracking design are insightful.

**Additional Feedback:**

For the experiment results in Table 4, please explain more clearly the difference between GGN+oracle and GGN+InstEmb.

**Documentation:**

Dataset download link is provided. The website for the presentation of the dataset can be made in a better way.

**Ethics:**

NA.

**Limitations:**

It might be better for the paper to give some suggestions on how to improve the re-detection performance for long-term egocentric visual tracking.

**Opportunities For Improvement:**

I don't find obvious drawback from this paper. One minor concern is about the stated baseline model of EgoSTARK. While there are experiments showing improvements made to the STARK tracker with finetuning and data augmentation, I don't see how exactly the EgoSTARK stands for.

**Relation To Prior Work:**

Yes.

**Summary And Contributions:**

This paper proposes a dataset for visual object tracking in egocentric videos (named as EgoTracks). The dataset is contructed based on videos from Ego4D dataset, and dense annotations are provided for the target object in the whole videos. Extensive experiments are conducted to evaluate the performance and analyze the drawbacks of state-of-the-art tracking algorithms on EgoTracks.

---

> ### Author Response · Authors · 2023-08-16
>
> We thank the reviewer for their positive feedback.
>
> EgoSTARK: We frame EgoSTARK as a strong baseline model for egocentric tracking applications, providing easy access to model and weights for off-the-shelf use. As we state in the paper, the nature of previous popular tracking datasets have led past trackers to adopt spatiotemporal priors that are overfit to these simpler settings; these cause trackers to do poorly on egocentric videos in ways that might not be obvious unless one thinks about how these trackers are engineered under the hood. We performed an empirical investigation into these assumptions, and used our learnings to produce a much stronger baseline model. We believe EgoSTARK will provide immediate value to practitioners in need for a strong, off-the-shelf tracker for egocentric video.
>
> GGN+oracle vs GGN+InstEmb: This was explained in Section B.1 + B.2 in the Supplementary Material, referenced on Line 261 of the main paper. We will expand upon this further in the next version of the paper. To recap, “Oracle” for tracking by detection refers to selecting the bounding box from the detector predictions with the highest overlap with the ground truth as the instance association method. The significant gap between these Oracle methods and the instance embedding method illustrates the major challenge of instance association for our dataset.
>
> We see a few promising directions for improving the re-detection performance for long-term egocentric visual tracking:
>
> a) Stronger features for associating objects should significantly improve re-detection; the impact of insufficiently discriminative feature embeddings can be clearly seen in the major gap in Tracking by Detection performance between the Oracle and InstEmb methods at the bottom of Table 4. Geometric keypoints, optical flow, or long-term trajectories [a] can also lead to large improvements here.
>
> b) Leveraging spatial signals: camera trajectories can be estimated as additional signals to the tracker. For example, if an object remains static during the window where it is out-of-view, knowledge of camera location can help re-localize the position of this object.
>
> c) Global, multi-view object representations: Egocentric videos, with their diverse camera trajectories and tendency to capture the camera wearer’s interactions with objects, often offer significantly richer and more varied viewpoints of objects than traditional third-person tracking datasets. In the latter, object appearances tend to be more constant, so modern tracking methods have thus far been able to get away with using a single image template (optionally with an additional template from the latest frame). With a need for more robustness to the different viewpoints and occlusions offered by egocentric video, we believe that a challenging egocentric tracking dataset like EgoTracks represents an opportunity to develop trackers with more global, view-variant object representations learned in an online fashion. A simple version of this can be found in Section D of the supplementary material, where we augmented EgoSTARK by fusing multiple templates; we found that such a strategy indeed improved tracking results on EgoTracks.
>
> [a] Wang, Qianqian, et al. "Tracking Everything Everywhere All at Once." arXiv preprint arXiv:2306.05422 (2023).

---

### Official Review · Reviewer_XGD2 · 2023-07-24

**Rating:** 6
**Confidence:** 4
**Correctness:** Yes
**Clarity:** Yes

**Strengths:**

Based on Ego4D, the scale of the dataset is an order of magnitude larger than existing ones. At the same time, the annotation effort is also huge. Really appreciate the hard work of the authors.

**Additional Feedback:**

N/A

**Documentation:**

Yes, included in Ego4D dataset

**Limitations:**

See above

**Opportunities For Improvement:**

1. I find the main issues of the dataset is the evaluation metric. The metrics proposed all focus on short-term aspect such as IoU. Since the characteristic of the dataset is long-term, it should have some metrics to reflect the performance of long-term tracking, such as the successful rate of re-detection, etc
2. Will the tracker be penalized if the objects disappear in certain frames but the tracker still outputs it? It is not clear how the certain metrics are computed.
3. Is there any statistics of the objects to be tracked, such as catagory or size?
4. The oracle setting in Table4 needs elaboration.
5. In the VCT setting, how the certain frame is chosen? How the performance varies given different choice of the initial frame?

**Relation To Prior Work:**

Yes

**Summary And Contributions:**

This paper presents a new egocentric tracking dataset. Different from previous work, the clips in this dataset is significantly longer than previous ones, and the egocentric nature makes the object motion is much larger than existing ones. Moreover, the objects may appear and disappear many times in one clip, making the dataset a challenging long-term tracking setting. The authors benchmark several SOTA methods and improve them by several parameter tuning and augmentations, which achieves better results.

---

> ### Author Response · Authors · 2023-08-16
>
> 1. Evaluation metric: Thanks for the suggestion. First, we observe that re-detection failures are indeed captured by our current reported metrics and cause significant drops in measured performance:
>
>     a) Recall (and therefore F-Score) is negatively impacted by failed re-detection, as the number of true positive frames will be 0 for the frames of the video where the object has been lost.
>
>     b) We evaluated trackers under an alternative evaluation protocol ($OFFT-\overrightarrow{OO}$) that removes the need for re-detection entirely. The significantly higher values in Table 5 (left) compared to Table 4 illustrate how much of a challenge re-detection is in our dataset.
>
>     At the reviewer’s suggestion though, we add additional results below to more directly capture re-detection rate. Specifically, we pose it as a per-occurrence detection problem, where we consider that the tracker has successfully re-detected an object during an occurrence if there is at least one frame during the occurrence where the tracker produced a bounding box above some IOU threshold with the ground truth. We report the average proportion of occurrences for which the tracker meets this criterion below, at two IOU thresholds: 0.5 and 0.75.
>
>     |                | Recall@IoU 0.5 | Recall@IoU 0.75 |
>     |----------------|----------------|-----------------|
>     | KYS            | 21.90          | 16.54           |
>     | DiMP           | 22.35          | 16.77           |
>     | GlobalTrack    | 42.72          | 35.55           |
>     | LTMU           | 41.18          | 34.67           |
>     | ToMP           | 39.85          | 34.81           |
>     | Siam-RCNN      | 47.73          | 44.70           |
>     | MixFormer      | 37.57          | 25.61           |
>     | STARK - Res50  | 43.31          | 39.00           |
>     | STARK - Res101 | 42.40          | 38.34           |
>     | EgoSTARK       | 66.83          | 59.95           |
>
>     The new experiment confirms that recall (F-score) is consistent with the success rate of re-detection: methods with higher recall (therefore F-Score) also have higher re-detection success rate. The experiment has been added as Supplementary Material Section E.
>
> 2. False positive penalty: Yes, a tracker that predicts object locations when the object has disappeared from view (i.e. a false positive) is penalized by both F-Score and Precision metrics included in almost all of our tables. The one exception is in Section 4.3, where we evaluate under the $OFFT-\overrightarrow{OO}$ protocol, which removes the frames where the object is absent from evaluation. We do this as an ablation study, to explicitly quantify the challenge of re-detection; as can be seen in Table 5, trackers will score higher when re-detection isn’t required.
>
> 3. Object statistics: Since we chose our objects directly from Ego4D’s Visual Query benchmark, we share the same object statistics for categories.
> A word cloud showing the diversity of object classes (and video scenarios) can be found in Section A.1 of the Supplementary Material. Further quantitative object class statistics can be found in the original Ego4D paper.
> Note that our tracklet attributes annotations (Section 3.3) do capture valuable traits (“active”, “transformed”, “recognizable”) that have a sizable impact on tracking performance (Table 6).
> At the reviewer’s suggestion, we also have plotted the distribution of object sizes relative to image sizes: $\sqrt{obj_w \times obj_h} / \sqrt{img_w \times img_h}$. Please see Supplementary Material Section A3 in the revision (also available from this link: https://ibb.co/SmJZRQF).
>
> 4. Oracle description: This was explained in Section B.1 (Lines 36-37) in the Supplementary Material, referenced on Line 261 of the main paper. We will expand upon this further in the next version of the paper. To recap, “Oracle” for tracking by detection refers to selecting the bounding box from the detector predictions with the highest overlap with the ground truth as the instance association method. The significant gap between these Oracle methods and the instance embedding method illustrates the major challenge of instance association for our dataset.
>
> 5. VCT setting: This setting refers to using the frame provided by Ego4D’s Visual Query benchmark as the template (see Section 3.1, line 212-214). We use this frame as a template for many of our experiments because by nature of their selection for Ego4D’s Visual Query benchmark, they should be good views of the object. Naturally, if the template frame is of poorer quality, we’d expect tracker performance to drop commensurately. Section C of the Supplementary Material demonstrated this empirically.

---

### Official Review · Reviewer_N2DY · 2023-07-27
**A long-term visual object tracking dataset from the egocentric perspective that is marginal above the acceptance threshold.**

**Rating:** 6
**Confidence:** 5
**Clarity:** The paper is well-written and easy to…

**Strengths:**

The dataset is huge and may help develop and evaluate new tracking algorithms.
The egocentric perspective provides challenging factors for object tracking.

**Additional Feedback:**

It is good to see such a large dataset constructed and released to the community. However, as a NeurIPS paper in the Datasets and Benchmarks track, the motivation and documentation of the dataset, as well as its potential help to the tracking community, should be highlighted.

**Correctness:**

The two main contributions claimed are supported in the manuscript. The construction process of the dataset is reasonable.

**Documentation:**

The documentation is lacking.

**Ethics:**

I do not see any ethical issues in the dataset.

**Limitations:**

The documentation part is weak.
The experimental analyses

**Opportunities For Improvement:**

The documentation work needs to be added to the dataset.
More rich annotations of the dataset, e.g., different kinds of challenging factors, will be more helpful for tracking algorithm development.
More deep analyses in the experimental part will also help.

**Relation To Prior Work:**

The relations to prior work and dataset are described in Section 2. The differences are well-discussed.

**Summary And Contributions:**

This work contributes a long-term visual object tracking dataset from the egocentric perspective. The constructed dataset contains more than 600 hours of videos with an average length of about 6 minutes. The paper also provides experimental results and improvements of the STARK tracker on the constructed dataset.

---

> ### Author Response · Authors · 2023-08-16
>
> Thank you for the feedback. We would find it helpful if you could provide more details for the opportunities for improvement.
>
> 1. Documentation: Besides the main paper itself, we provided significant additional description in the Supplementary Materials, including an extensive data sheet. We also included documentation and commands for downloading, processing, and training on EgoTracks in our GitHub repository: https://github.com/EGO4D/episodic-memory/tree/main/EgoTracks. We note that other reviewers found our documentation well-written (Pzo5, G7Uz).
>
> 2. Additional annotations: We believe the large size (“huge”, as the reviewer puts it) of the dataset in terms of total video hours, average length, and number bounding boxes and the challenging and underexplored egocentric setting should already be a sufficient contribution. We also point out that we did include additional attribute annotations capturing when objects are being used by the camera wearer (resulting in hand occlusions and rapid motion), or when objects are transformed (appearance changes); see Section 3.3, Figure 3, and Table 2. In Table 6, we empirically demonstrate that these attributes do capture tracklets that are especially difficult for object trackers. Finally, we’d like to highlight that the additional evaluation protocols proposed in the paper (e.g. $OFFT-\overrightarrow{OO}$, see Sec 4.1) are specifically designed to tease apart the various challenges of our dataset.
>
> 3. Experimental Analysis: We want to highlight that the other reviewers reacted positively to our “extensive” (hGyN, G7Uz), ”comprehensive” (Pzo5), and “insightful” (hGyN) experiments. If the reviewer has any specific experimental analysis that the reviewer would find helpful, we would be happy to do our best to accommodate.

---

### Author Response · Authors · 2023-08-16

We thank all the reviewers for their helpful comments. Summarizing some common positive themes:
* We are glad to know that the reviewers all agree on the value of our dataset, in particular its “huge” size (N2DY, XGD2) and differentiating characteristics that offer new challenges beyond existing tracking datasets (XGD2, Pzo5, G7Uz).
* Beyond the data itself, our “extensive” (hGyN, G7Uz), ”comprehensive” (Pzo5), and “insightful” (hGyN) experiments were also viewed positively as contributing to understanding how trackers perform and the unique challenges of our dataset.
* Reviewers also found our paper well-written (N2DY, hGyN, Pzo5, G7Uz).

We provide responses to individual comments or questions below each review directly. We also post a revision of the paper with changes highlighted in red, incorporating reviewers’ suggestions.

---

### Author Response · Authors · 2023-08-25

We sincerely thank the reviewers again for their helpful suggestions and insightful feedback. We hope our revision properly addresses reviewers’ concerns. We would like to check if there are additional concerns or comments, and we are happy to provide further discussions, clarity or ablations.

---

### Decision · Program_Chairs · 2023-09-22

**Decision:**

Accept (Poster)

**Comment:**

The paper presents a dataset for long-term egocentric visual object tracking EgoTracks. While there have been many datasets and challenges organised around visual object tracking from the third-person perspective, the specific challenges facing long-term egocentric visual object tracking have been underrepresented and this paper aims at filling this gap. In particular it demonstrates the challenges caused by disappearance and re-appearance of the target objects, large camera motions and hand interactions. In addition, the paper contains extensive experiments to demonstrate the performance of the current deep trackers on this new dataset and propose how to make improvements for the egocentric data. The initial assessments of the quality of the paper by all five reviewers have been positive, stressing the importance, size and quality of the dataset. They have also appreciated the extensive and insightful experiments. Also, clarity and correctness have been assessed positively. No ethical concerns have been raised. However, the reviewers posed a few questions that required clarifications (e.g., related to the evaluation matric) and proposed a number of potential improvements and additional experiments. They also requested the authors to comment more in-depth on the future directions. In response, the authors provided the requested clarifications and performed additional experiments (e.g., performance of the EgoSTARK on other datasets). In summary, at the end of this process, all the reviewers indicated that the paper should be accepted.